# Preparation and Ballistic Performance of a Multi-Layer Armor System Composed of Kevlar/Polyurea Composites and Shear Thickening Fluid (STF)-Filled Paper Honeycomb Panels

**DOI:** 10.3390/polym13183080

**Published:** 2021-09-13

**Authors:** Chang-Pin Chang, Cheng-Hung Shih, Jhu-Lin You, Meng-Jey Youh, Yih-Ming Liu, Ming-Der Ger

**Affiliations:** 1Department of Chemical & Materials Engineering, Chung Cheng Institute of Technology, National Defense University, Taoyuan 335, Taiwan; changpin24@gmail.com (C.-P.C.); yolin1014001@gmail.com (J.-L.Y.); takuluLiu@gmail.com (Y.-M.L.); 2System Engineering and Technology Program, National Chiao Tung University, Hsinchu 300, Taiwan; 3Graduate School of Defense Science, Chung Cheng Institute of Technology, National Defense University, Taoyuan 335, Taiwan; s922326@gmail.com; 4Department of Mechanical Engineering, Ming Chi University of Technology, Taishan, New Taipei City 243, Taiwan

**Keywords:** multi-layer armor, ballistic performance, polyurea elastomers, shear thickening fluid

## Abstract

In this study, the ballistic performance of armors composed of a polyurea elastomer/Kevlar fabric composite and a shear thickening fluid (STF) structure was investigated. The polyurea used was a reaction product of aromatic diphenylmethane isocyanate (A agent) and amine-terminated polyether resin (B agent). The A and B agents were diluted, mixed and brushed onto Kevlar fabric. After the reaction of A and B agents was complete, the polyurea/Kevlar composite was formed. STF structure was prepared through pouring the STF into a honeycomb paper panel. The ballistic tests were conducted with reference to NIJ 0101.06 Ballistic Test Specification Class II and Class IIIA, using 9 mm FMJ and 44 magnum bullets. The ballistic test results reveal that polyurea/Kevlar fabric composites offer better impact resistance than conventional Kevlar fabrics and a 2 mm STF structure could replace approximately 10 layers of Kevlar in a ballistic resistant layer. Our results also showed that a high-strength composite laminate using the best polyurea/Kevlar plates combined with the STF structure was more than 17% lighter and thinner than the conventional Kevlar laminate, indicating that the high-strength protective material developed in this study is superior to the traditional protective materials.

## 1. Introduction

With the development of technology and techniques, the structural design of armor systems is gradually moving towards a multi-layer armor system [1,2,3]. For protection purposes, the multi-layer armor system can be used as a simple way of increasing the number of layers of protection material to meet the need for increased impact resistance and to reduce the additional costs associated with engineering changes. However, this approach also tends to increase the thickness and weight of the garment, causing restrictions on movement and strain on the wearer, so the light weighting of protective equipment has been a continuous improvement effort by developers from all walks of life [4,5,6]. Bulletproof materials must have high strength and modulus to protect against shear damage and tensile deformation caused by projectiles during high-speed impacts [7,8,9]. The mass effect is an important factor affecting impact resistance, and in the pursuit of light weight, it also affects the protection capability of ballistic materials. Recently, many studies have proposed the use of composite materials to achieve the goal of improving impact resistance [10,11,12]. In addition, shear thickened fluids have been subjected to a series of tests in the field of impact resistant materials in recent years due to their special shear thickening properties.

Shear thickened fluids are intelligent materials that are non-Newtonian in nature and exhibit solid-like properties when the shear force or shear rate exceeds the critical shear stress or critical shear rate, resulting in a rapid increase in viscosity [13,14,15,16,17,18]. Due to this unique shear thickening rheological property, they facilitate the absorption of impact energy for protection in the event of an impact. However, in a study by Arora et al., it was noted that increased yarn-to-yarn friction was not necessarily beneficial in terms of absorbing impact energy, but rather reduced the protective capacity of these high-performance fabrics made from finer yarns shaped from specific fabrics [19]. The results of these studies have also highlighted a key point that when combining STF with other materials for impact-resistant protective gear applications, the mutual influence between the rheological properties of STF and the structural parameters of the combined material must be taken into account, in order to avoid stress concentration in the composite material during the impact process, which could lead to a weakness in the material and loss of the original protective effect, which plays a crucial role in determining its impact resistance.

Reviewing the literature and fundamental theories on the ballistic mechanism of high-performance Kevlar fiber fabric, the invasion process and damage mechanism of high-performance Kevlar fiber fabric subjected to high-velocity projectile impact are well understood [20,21,22,23,24]. A large number of studies have now shown that 2D woven fabrics can be coated with polymers, for example, to improve the fracture toughness of the structure [25,26,27,28,29]. In this study, an attempt was made to enhance the impact resistance of high-performance Kevlar fabric with polyurea polymers. From the literature on high-performance Kevlar fiber composites, it is clear that when adding reinforcement to high-performance Kevlar fiber fabrics, attention must be paid to the effect of the added content on their ballistic protection mechanism [4,30,31]. Therefore, in this study, polyurea elastomer/Kevlar composites were prepared using diluted tetrahydrofuran solution containing various ratios of polyurea polymer raw materials in order to obtain suitable composites.

The multi-layer armor system blunts the penetration and penetration capability of high-speed projectiles by means of a protective material at the front end of the armor stack, and then depletes the kinetic energy of the warhead by means of a bulletproof material at the rear end. In addition to the unique properties of the material (fibers) itself, a variety of mechanisms should be considered to enhance impact resistance [32,33,34,35]. For example, after selecting the appropriate material properties, a common way to improve protection is to add more layers in a different arrangement during the production of the laminate stack. However, even though increasing the number of layers in a composite material stack can improve protection, it can also affect the total weight and flexibility of the final protective equipment [36,37,38,39]. In order to achieve the goal of developing a soft armor, the multi-layer armor designed in the study uses fibers as the main structure at the front end and incorporates an STF structure at the rear end to enhance the efficiency of impact energy absorption.

This research follows our previous research on high energy absorbing nano-mesophase reinforced ballistic materials. Using the experience in developing ballistic materials, we further combine the characteristics of polymer composite materials and STF structures, with the aim to produce high-strength composite ballistic material with excellent protection for users. It has been reported that polymer matric composites or polymer-coated fabrics can change their ballistic performance [30,40]. However, the impact behavior and failure mode are affected by the polymer used [41,42]. Due to its effective energy absorption property, polyurea is commonly used as a protective coating on concrete or steel structures. However, studies on the ballistic performance of polyurea-coated fabric are rare [43]. For this purpose, high-velocity impact tests were conducted according to the NIJ 0101.06 Class IIIA standard [44] to determine the ballistic performance of neat Kevlar fabric and samples composed of Kevlar/polyurea composites and STF structures. The remarkable results from the comparison between the untreated Kevlar fabrics and panels with an adequate arrangement of Kevlar/polyurea composites and STF structure showed that a high-strength composite laminate using the best polyurea/Kevlar plates combined with the STF structure was lighter and thinner than the conventional Kevlar laminate, while maintaining the same ballistic resistance.

## 2. Experimental

### 2.1. Materials

In this study, nanoscale silica was mixed with polyethylene glycol (PEG) to prepare a shear thickening fluid. The silica was a nanoscale fumed silica powder and purchased from Kingmaker Chemical (ECHO, Miaoli, Taiwan). PEG was an oligomer produced by ECHO with a molecular weight of 200 g/mol. A wet dispersant, Disperbyk-111, was used to enhance the stability of silica particles.

### 2.2. Fabrication of STF-Filled Paper Honeycomb

When preparing STF, problems such as uneven mixing are often encountered. Therefore, in the preparation process, we initially used a planetary mixer to mix the solid dispersed particles with the liquid dispersing medium, and then used a three-roller mixer to roll the shear thickening fluid to disperse the solid particles in the shear thickening fluid well. In the first stage, the required amount of polyethylene glycol was weighed with an electronic scale and poured into the mixing tank of the planetary mixer, and the required amount of nanosilica (10–30 nm particle size) and additive (Disperbyk-111) was weighed and poured into the tank. The mixing tank was then placed in the planetary mixer and the suspension was stirred at speed of 2000 rpm for 10 min. After that, the suspension was defoamed at 2500 rpm for 5 min. The above steps were repeated until the silica content of the shear thickening liquid reached the required solid content. In the second stage, the shear thickening liquid prepared in the first stage was fed into and passed three roll mills five times with a gap of 150, 100, 50, 10 and 5 μm, separately, to improve the homogeneity of the STF. As shown in Figure 1, the STF-filled paper honeycomb structure was prepared through pouring the STF into a honeycomb paper panel (Asazawa Industrial Co., Ltd., Taoyuan, Taiwan) with a specified size and an NY vacuum bag, provided by Futian Packaging Co., Ltd, Taipei, Taiwan was used to seal the STF. The purpose of using honeycomb paper spacers was to maintain the shear thickened liquid in a liquid state. By controlling the size of the honeycomb spacer, the amount of shear thickened liquid can be fixed and the thickness as well as weight of the target plate can be controlled.

### 2.3. Preparation of Polyurea Elastomer/Kevlar Plates

The preparation process is displayed in Figure 2. The sample layers were finally investigated according to the subsequent experimental design.

### 2.4. Rheological Test

Rheological measurements of STF suspensions were performed at 25 °C using a stress-controlled rheometer (model: HAAKE RS600, Thermo Fisher Scientific, Newington, CT, USA). For rheometer testing, a tapered plate (No. C20/2 Ti) with a flat fixture was used for measurement at room temperature (25 °C). The diameter of the tapered plate was 20 mm, the angle of the outer taper was 2 degrees and the distance between measurement positions (gap) was set at 0.1 mm.

### 2.5. Stab Tests

A home-made drop hammer tester (Figure 3) was used to perform the puncture resistance test. The test was carried out with a fixed weight of 176.3 g and a knife weighing 102.6 g, for a total weight of 278.9 g and a height of 142 cm. The puncture effect was assessed by impacting the piercing knife on a high energy absorbing nano-media reinforced ballistic material on the test bench using the free fall principle with a certain amount of energy.

### 2.6. Ballistic Impact Testing

The ballistic impact tests were conducted with reference to NIJ 0101.06 Ballistic Test Specification Class II and Class IIIA [44], using 9 mm FMJ and 44 magnum bullets for live ballistic testing. Ballistic tests were conducted with the projectile flying in a steady direction through the inner rifling of the accuracy barrel to the target, with the muzzle 5 m from the target and the target 6 m from the wooden retaining wall. A schematic diagram of the ballistic test is shown in Figure 4. The first set of light gates was set at the muzzle from the target to measure the initial velocity of the projectile, and the second set of light gates was set from the target to the wooden retaining wall to measure the final velocity of the projectile, and the depth of the rear mud depression was the standard for impact resistance. Although a bulletproof vest is generally resistant to penetration by bullets, it can also cause bodily injury due to the force of the impact, so the depth of indentation in the ballistic test has always been a key criterion for a bulletproof vest. In this study, the back face signature (BFS), as defined by the NIJ 0101.06 standard, was used to determine the level of protection of the protective material and to compare the protection performance of the samples in terms of the depth of depression. In addition, during the ballistic test, we used Photron’s FASTCAM SA1.1 high-speed video camera to capture the dynamic changes in the STF structure during the impact of the bullet on the bullet-resistant laminated sample.

## 3. Results and Discussion

### 3.1. Rheological Analysis of the STF

By measuring the rheological properties and analyzing the rheology, we could further identify the physical changes in the flow and deformation of the shear thickened liquid materials prepared for the study. The solid content of STF is an important parameter affecting the rheological properties of STF. According to the literature, as the solid content of STF increases, the shear thickening properties of STF will reach the maximum viscosity more rapidly with the increase in shear rate, and STF with a higher solid content has a higher maximum viscosity value [45]. By controlling the solid content of the STF, the same can be obtained for STF materials with different maximum viscosities. Figure 5a shows the rheological curves obtained from the planetary mixer and the triple-roller blender after processing different STF solid contents by rheological analysis. From Figure 5a, it can be seen that as the solid content of STF increases, the maximum viscosity obtained by the STF material under shear also tends to increase. In order to achieve the level of ballistic protection and high-velocity impact resistance required for practical protection applications, it is essential that the solid particles in the STF should be evenly dispersed in the solvent. Another important point to note is the reversibility of STF. In protective applications, STF differs from other materials in that it hardens on impact, but slowly returns to its fluid state after the impact force has dissipated. This special phenomenon also allows the use of STF in bulletproof vests to avoid “secondary damage” during the bulletproofing process compared to rigid vests. Therefore, in addition to the use of a triple-roller mixer to improve the dispersibility of the STF, the addition of an interfacial activator to the system to promote the homogeneous dispersion of the particles in the suspension was used to further increase the dispersibility, reversibility and stability of the prepared STF system. The main function of adding an interfacial activator is to reduce the interfacial tension between SiO_2_ and PEG (solid–liquid) in the system and to help prevent the coalescence of SiO_2_ particles. The addition of a dispersant has the effect of maintaining the stability of the dispersion in the normal fluid state of the STF, and also helps with the reversibility of the STF after it has been impacted and formed a solid state, reducing the time required to return to the original fluid state. Some of the test results on the effect of the added dispersant content on the rheological properties of the STF are shown in Figure 5b. The results of Figure 5b show that the rheological curve of STF gradually shifts to the left and the critical shear rate gradually decreases with the increase in the added dispersant. The trend of the experimental results obtained is consistent with the results in the literature on the effect of different molecular weights of liquid media on STF [46]. The addition of dispersant increases the molecular weight of the liquid medium in the STF, and as the molecular weight of the liquid medium increases, the viscosity of the liquid medium also increases. However, as the dispersant affects the coalescence of the solid particles in the STF, the observed shear thickening curve tends to flatten out as the dispersant content in the STF increases. In order to reduce the effect of dispersant on the rheological properties of STF and to increase the dispersion effect of STF, a dispersant amount of 3 g was used in this study for subsequent experiments.

### 3.2. Ballistic Performance of Kevlar/STF-Filled Paper Honeycomb Plates

The rheological properties of STFs required to resist high- or low-velocity impacts are very different, so STFs with different rheological properties should be used for different types of protection in order to achieve efficient protection capabilities. Our previous studies have shown that STF with a high critical shear rate has better protection properties [47]. A shear thickening fluid with a solid content of 40 wt% was prepared using nano-grade silica and poured into honeycomb paper of various thicknesses to continue the research on weight reduction and thinning of ballistic materials to find a balance between weight, thickness and protection. The test specimens were 40 wt% solids shear thickening fluid in honeycomb paper of 2 mm, 3 mm and 4 mm in thickness, sealed with PE bags and combined with Kevlar. Ballistic testing has been performed on different laminating sequences of composite panels composed of one STF structure layer and nineteen layers of Kevlar fabric which were obtained by simply placing the STF structure at different positions in the composite panels in our previous study [47] and showed that the STF structure placed at the rear position can significantly contribute to the increase in impact resistance. Thus, the test specimens used in the following study were prepared by putting Kevlar laminate in the front end and the STF structure in the rear end. Ballistic testing was conducted using National 9 mm pistol ammunition and 44 Magnum pistol ammunition in accordance with NIJ 0101.06 Class IIA. As can be seen from the test results in Table 1, compared to the control 29-layer Kevlar sample (Std. 29), the A-1 sample prepared in the first stage of research was already lighter and thinner than the control 29-layer Kevlar sample, and a comparison of the sludge depression depth results between the two indicated that the STF/Kevlar composite (No. A-1) prepared was more impact resistant. This result also shows that the amount of shear thickening fluid in the sample is directly proportional to the impact resistance, and that shear thickening fluid is indeed effective in absorbing impact energy. Based on the results of these experiments, the design of a bulletproof vest can be further developed in accordance with the product specifications to find the optimum weight and thickness of the bullet-resistant structure. In addition, for the ballistic test upgraded to NIJ 0101.06 Level IIIA, the test results are shown in Table 1, for samples A-4 and A-5. The test results show that a stack of 37 layers of Kevlar fabric and an STF structure positioned at the second layer is needed to satisfy the requirement of NIJ 0101.06 Level IIIA.

### 3.3. Ballistic Performance of the Polyurea Elastomer/Kevlar Plates

From reviewing the literature and fundamental theories on the ballistic protection mechanism of high-performance Kevlar fiber fabric, the process and damage mechanism of high-performance Kevlar fiber fabric exposed to high-velocity projectile impact are well understood. Using the results of previous experiments on the impact resistance of high-performance fiber composites [48], Kevlar high-performance fiber fabric coated with polyurea was used to enhance the impact resistance. Then, a comparison of the impact resistance between the Kevlar fabric coated with polyurea and STF-impregnated Kevlar fiber composites reported in the literature was made. From the literature on Kevlar fiber composites, it is clear that when reinforcements are added into Kevlar fiber fabrics, attention must be paid to the effect of the amount of reinforcement added on their ballistic protection mechanism. The test results (Table 2) show that, when polyurea elastomers diluted with THF are brushed onto Kevlar, the performance of the Kevlar fabric is improved and the number of layers of the Kevlar stack is reduced, and Class II ballistic testing shows that there is one level above the original Class IIA in ballistic performance. When a projectile impacts a fabric, it is considered that the projectile kinetic energy is dissipated through a combination of mechanisms such as tension in primary yarns, deformation of fabric, energy dissipated through frictional slips (yarn/yarn and projectile/yarn), yarn breakage and yarn pull-out from the fabric [49]. The polyurea coating helps primary yarns to transfer the impact load into the secondary yarns and it also increases the friction between yarns. Consequently, the Kevlar/polyurea composites achieved a better ballistic performance compared to the pristine Kevlar fabric. The results from Table 2 for No. B-3 and No. B-4 tests showed that they were approximately the same, so we chose the lighter sample parameters and used a 1:5 THF dilution ratio for the following tests.

### 3.4. Stab Resistance of the Polyurea Elastomer/Kevlar Plates

Commercially available Kevlar fabric is woven in an orthogonal plain weave, which makes it difficult to protect against the impact of sharp projectiles. The puncture resistance of the polyurea elastomer/Kevlar composite was verified by simulating a bayonet puncture attack using a drop hammer puncture tester and comparing it with the Kevlar fabric as a control group. The results are presented in Table 3. As can be seen from Table 3 for Test No. D-1, a combined stack of 21 layers of Kevlar fabric is required to protect against the impact of a hammer piercing without penetration by a bayonet. However, a polyurea elastomer/Kevlar composite laminated construction (No. D-2) requires only 12 layers to resist the impact of a falling bayonet puncture without penetration when tested with the same standard drop hammer puncture test.

### 3.5. Ballistic Test Results of the Multi-Layer Armor System

For stacked structures, in order to enhance the protection performance of the ballistic stack, the number of stacked layers is generally increased in order to increase the surface density of the stacked structure to achieve the purpose of enhancing the protection performance. However, by increasing the number of layers, the weight of the protective equipment is increased and the load on the person wearing it is increased [50]. The variety of high-strength composites produced in this study provides more scope for design discretion in optimizing the impact resistance of the laminated structure due to the diverse properties of the composites. In order to achieve the goal of lightweighting of protective equipment, the experiments in this section were carried out to verify the design patterns of different functional composite materials and boundary conditions (target size and thickness of the STF structure) in order to obtain the optimal design pattern of the high-strength composite material stack.

To summarize the results of this study and to consider the need for a soft ballistic material, the final high-strength composite layer will be designed using the best impact-resistant high-performance polyurea elastomer/Kevlar fiber composite, combined with a flexible STF structure. The ballistic test was conducted under the conditions of NIJ 0101.06 Class IIIA (two levels higher than Class IIA), and the size of the sample prepared for the test was 13 cm × 13 cm, a smaller surface area than that of a normal bulletproof vest. A domestic soft bulletproof vest with 29 layers of Kevlar fabric was used for comparison. Although the sample had the same number of layers (29), it was not able to protect against Class IIIA projectiles. This could be attributed to the size boundary effect caused by its smaller dimensional area. Under Class IIIA test conditions, a 47-ply Kevlar fabric (No. Std. 47) was selected as the control group to compare the impact resistance of the experimental groups, which met the 44 mm depth of sludge depression. From the analysis of the above experimental results, a lighter polyurea elastomer/Kevlar laminate was used to replace the Kevlar fabric laminate structure at the front of the original sample, and a shear thickening fluid structure was added to form a high-strength composite ballistic laminate sample. Table 4 displays the results of ballistic tests which were conducted using the Class IIIA standard of NIJ 0101.06. It can be seen from Table 4 that the depth of sludge depressions for samples E-1 and E-6 is 38.01 mm and 43.38 mm, respectively. Both met the Class IIIA standard of NIJ 0101.06. The lighter and thinner sample, No. E-6, is 17.91% thinner and 17.68% lighter than the control group, No. Std. 47, indicating that the reinforced high-strength composite material is lighter and thinner and meets the Class IIIA standard of NIJ 0101.06. To optimize the high-strength composite laminates, samples E-1 and E-6 were prepared using the best data from this study and their ballistic tests were performed according to the Class IIIA standard of NIJ 0101.06. The depths of sludge depressions for these two samples were 40.01 mm and 43.38 mm, respectively, which met the requirements of the standard of less than 44.0 mm. The error value is approximately ±2.0 mm, which is in line with the standard error value set by the American Judicial Association.

## 4. Conclusions

In this study, we introduced a new approach for developing a high-strength composite laminate which is lighter and thinner than the conventional Kevlar laminate and can still meet the Class IIIA standard of NIJ 010106 for bulletproof vests. The results of this study show that relying only on shear thickening fluids to improve the protection of the elastic cascade is limited. On the other hand, it generally will increase the weight and thickness of the sample. In this study, a polyurea/Kevlar fabric composite was prepared and utilized to develop a protective material that is comfortable to wear and meets the requirements of reduced wear load. A close comparison in ballistic test results reveals that polyurea/Kevlar fabric composites offer better impact resistance than conventional Kevlar fabric. In order to further improve the protection performance of the body armor and to understand the correct application of STF, this study used STF structures to replace some of the layers of fabric in soft body armor. The results of the NIJ 0101.06 Class IIIA ballistic test showed that a 2 mm STF structure could replace approximately 10 layers of Kevlar in a ballistic-resistant layer. Finally, a high-strength composite laminate (13 cm × 13 cm) using the best polyurea/Kevlar plates combined with the STF structure was more than 17% lighter and thinner than the conventional Kevlar laminate. Further work needs to be carried out to obtain the optimal results. However, our research provides a promising way to fabricate protective materials that can be applied in the future.

## Figures and Tables

**Figure 1 polymers-13-03080-f001:**
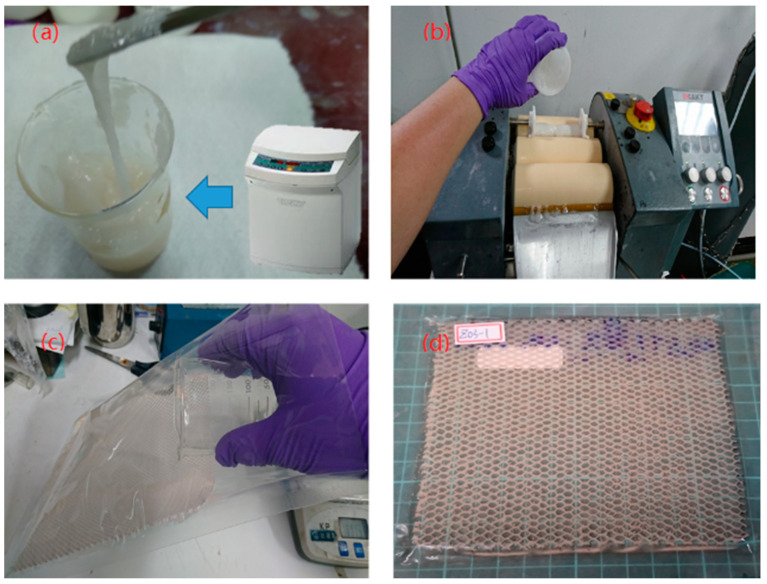
Fabrication of STF-filled paper honeycomb. (**a**) planetary mixer (**b**) three-roller mixer (**c**) Filled with good flowing STF into honeycomb paper (**d**) Schematic diagram of STF structure.

**Figure 2 polymers-13-03080-f002:**
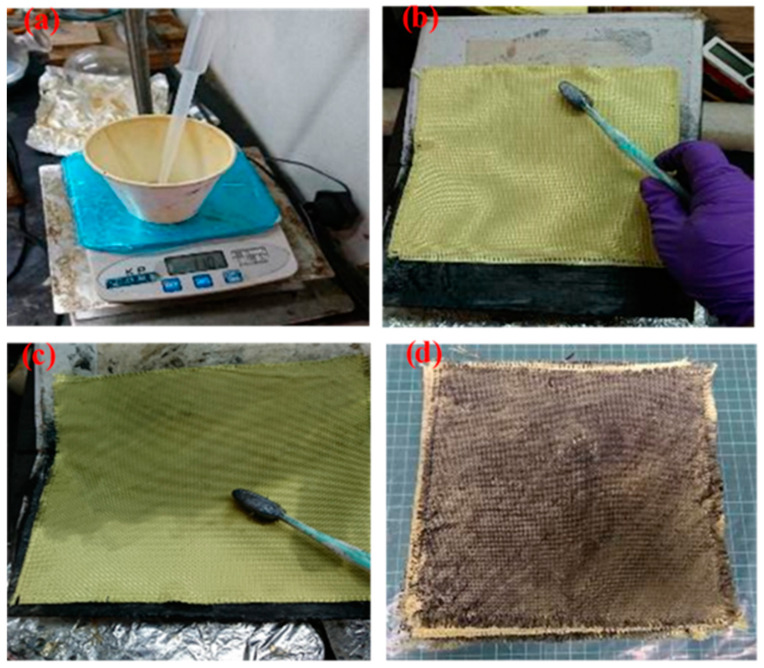
Preparation of polyurea elastomers/Kevlar plates (**a**) Preparation of polyurea elastomers (**b**) lay the cut Kevlar fabric flat on the carrier (**c**) painting polyurea elastomers (**d**) polyurea elastomers/Kevlar plates schematic diagram.

**Figure 3 polymers-13-03080-f003:**
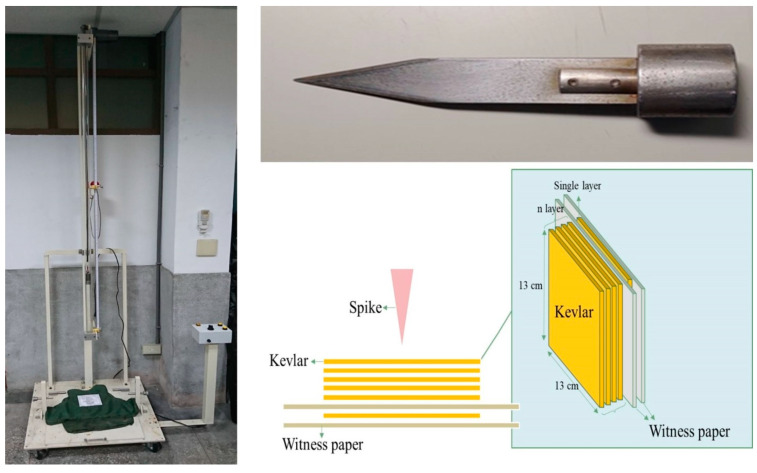
Schematic diagram of the stab test setup.

**Figure 4 polymers-13-03080-f004:**
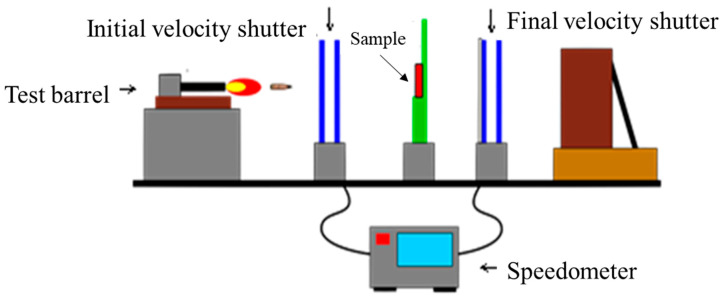
Schematic diagram of the ballistic test setup.

**Figure 5 polymers-13-03080-f005:**
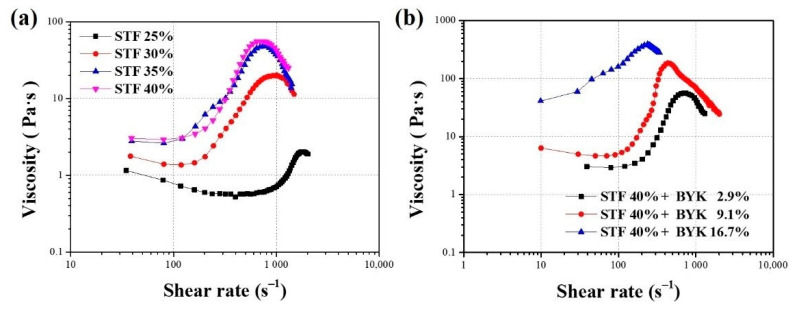
Rheology graph for as-prepared STF sample (**a**) STF rheological curves with different solid contents (**b**) The rheological curve of STF with 40% solid content and different content of dispersant.

**Table 1 polymers-13-03080-t001:** Results for ballistic test of Kevlar/STF-filled paper honeycomb plates.

Plate ID	Composition	Layers	Honeycomb Structure Thickness (mm)	Areal Density (g/cm^2^)	BFS (mm)	NIJ 0101.06 Test Level
Std. 29	29 plies of Kevlar fabric	29	-	1.29	34.01	ⅡA
A-1	18 plies of Kevlar fabric and one ply of STF-filled paper honeycomb structure	19	4	1.27	33.84	ⅡA
A-2		19	3	1.16	34.32	ⅡA
A-3		19	2	1.08	41.37	ⅡA
A-4	37 plies of Kevlar fabric and one ply of STF-filled paper honeycomb structure	38	4	2.08	39.45	IIIA
A-5		38	2	1.85	43.71	IIIA

**Table 2 polymers-13-03080-t002:** Results for ballistic test of polyurea elastomer/Kevlar plates.

Plate ID	Composition	Layers	Dilution Ratio (Polyurea/THF)	Areal Density (g/cm^2^)	BFS (mm)	NIJ 0101.06 Test Level
Std. 19	19 plies of Kevlar fabric	19	-	0.87	43.9	Ⅱ
B-1	19 plies of polyurea elastomer/ Kevlar plates	17	1:10	0.79	55.3	Ⅱ
B-2		17	1:5	0.83	36.2	Ⅱ
B-3		15	1:5	0.75	45.2	Ⅱ
B-4		15	1:3	0.91	46.4	Ⅱ
B-5		15	1:1	0.97	39.9	Ⅱ

**Table 3 polymers-13-03080-t003:** Results for stab tests.

Plate ID	Composition	Not Penetrated Limit (layers)	Weight (g)	Thickness (mm)
D-1	Neat Kevlar fabrics	21	150.5	11.17
D-2	Polyurea elastomers/ Kevlar	12	93.7	7.15

**Table 4 polymers-13-03080-t004:** The influence of boundary effects (target area size, surface density and thickness parameters).

Plate ID	Composition	Layers	Thickness (mm)	Areal Density (g/cm^2^)	BFS (mm)	NIJ 0101.06 Test Level
Std. 47	47 plies of Kevlar fabric	47	24.91	2.11	43.79	IIIA
E-1	Polyurea elastomer/ Kevlar plates (37-ply) + STF structure (40%, 2 mm, 1-ply)	38	23.22	1.98	38.01	IIIA
E-2	Polyurea elastomers Kevlar plates (35-ply) + STF structure (40%, 2 mm, 1-ply)	36	22.74	1.94	38.32	IIIA
E-3	Polyurea elastomer/ Kevlar plates (32-ply) + STF structure (40%, 2 mm, 1-ply)	33	20.31	1.77	40.01	IIIA
E-4	Polyurea elastomer/ Kevlar plates (32-ply) + STF structure (40%, 4 mm, 1-ply)	33	23.14	2.03	38.56	IIIA
E-5	Polyurea elastomer/ Kevlar plates (32-ply) + STF structure (40%, 3 mm, 1-ply)	33	22.12	1.91	37.58	IIIA
E-6	Polyurea elastomer/ Kevlar plates (32-ply) + STF structure (40%, 2 mm, 1-ply)	33	20.45	1.74	43.38	IIIA

## Data Availability

The data presented in this study are available on request from the corresponding authors.

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
