# Peer review of "Preparation and Ballistic Performance of a Multi-Layer Armor System Composed of Kevlar/Polyurea Composites and Shear Thickening Fluid (STF)-Filled Paper Honeycomb Panels"

_polymers, 2021, doi:10.3390/polym13183080_

Round 1

Reviewer 1 Report

Comments

This paper studied a multi-layer armor system composed of polyurea elastomer/aramid composites and shear thickening fluids (STFs) filled paper honeycomb panels. The outcome of the paper is interesting however, there are several aspects that need to be improved. The reviewer can only recommend for publication if the author satisfactorily address the following major comments in the revised version.

  1. Suggest to highlight key findings in the first dot point of the conclusion.
  2. The research questions and justification of selected parameters should be highlighted.
  3. The novelty of the study should be highlighted more clearly at the end of introduction section. How this study is different from the published study in literature?
  4. How the outcome of this study will benefit researchers and end users? This need to be highlighted in introduction or end of conclusion.
  5. The importance of dynamic loading on the fibre composites and the recent investigation should be discussed in introduction section to improve the background study. Recently, FRP composites were investigated under impact [Ref: Behaviour of continuous fibre composite sandwich core under low-velocity impact] and fatigue [Ref: Tensile fatigue behavior of polyester and vinyl ester based GFRP laminates—A comparative evaluation]. Suggest to include them in introduction section with proper citations to improve the background study.

I would be happy to see the revised version to understand how these comments are being addressed.

Reviewer 2 Report

The paper presents a series of information on the ballistic performance of composite materials.

- The paper presents a series of results that may be of interest to the scientific community:

- The title of the paper does not correspond to the content. Thus, in the title the authors talk about aramid composites and in the paper they use kevlar. Under these conditions, a correlation of the title of the paper with its content is required;

- Otherwise, the entire content of the paper must be rewritten because it is not in accordance with its title;

Given the above, the paper does not meet the criteria for publication in the journal Polymers, but I can make a number of recommendations on the structure of the paper:

- the research methodology needs to be substantially improved in the sense that a clear presentation of the structure of all types of tests performed is required. Also, in order to be able to draw a series of conclusions, the ballistic behavior of composite materials must be analyzed according to its structure;

- the scientific part cannot be identified in the paper;

- it is necessary to present macroscopic images with the specimens made before and after the ballistic test:

- the discussion part needs to be substantially improved;

- the analysis of the results presented in the paper is one more quantitative; a qualitative analysis must be performed;

- the resolution of some figures needs to be improved;

- the conclusions should include future research directions;

- the bibliography does not respect the format imposed by the journal;

Round 2

Reviewer 1 Report

I have no further comments.

Reviewer 2 Report

The authors revised their manuscript according to my suggestions. Thus the manuscript can be accepted for publication.